microbiology

harbour porpoise, grey seal, common seal, microbiome, bacterial transmission

**Author for correspondence:**
Jaap A. Wagenaar
e-mail: j.wagenaar@uu.nl

# After the bite: bacterial transmission from grey seals (*Halichoerus grypus*) to harbour porpoises (*Phocoena phocoena*)

Maarten J. Gilbert[1,3], Lonneke L. IJsseldijk[2],
Ana Rubio-García[1,4,5], Andrea Gröne[2], Birgitta Duim[1,6],
John Rossen[5], Aldert L. Zomer[1,6]
and Jaap A. Wagenaar[1,6,7]

[1]Faculty of Veterinary Medicine, Department of Infectious Diseases and Immunology, and
[2]Faculty of Veterinary Medicine, Department of Biomolecular Health Sciences, Division of Pathology, Utrecht University, Utrecht, The Netherlands
[3]Reptile, Amphibian and Fish Conservation Netherlands (RAVON), Nijmegen, The Netherlands
[4]Sealcentre, Pieterburen, The Netherlands
[5]Department of Medical Microbiology and Infection Prevention, University of Groningen, University Medical Center Groningen, Groningen, The Netherlands
[6]WHO Collaborating Centre for Campylobacter/OIE Reference Laboratory for Campylobacteriosis, Utrecht, The Netherlands
[7]Wageningen Bioveterinary Research, Lelystad, The Netherlands

MJG, 0000-0002-9967-2936; JR, 0000-0002-7167-8623

Recent population growth of the harbour porpoise (*Phocoena phocoena*), grey seal (*Halichoerus grypus*) and common seal (*Phoca vitulina*) in the North Sea has increased potential interaction between these species. Grey seals are known to attack harbour porpoises. Some harbour porpoises survive initially, but succumb eventually, often showing severely infected skin lesions. Bacteria transferred from the grey seal oral cavity may be involved in these infections and eventual death of the animal. In humans, seal bites are known to cause severe infections. In this study, a 16S rRNA-based microbiome sequencing approach is used to identify the oral bacterial diversity in harbour porpoises, grey seals and common seals; detect the potential transfer of bacteria from grey seals to harbour porpoises by biting and provide insights in the bacteria with zoonotic potential present in the seal oral cavity. β-diversity analysis showed that 12.9% (4/31) of the harbour porpoise skin lesion microbiomes resembled seal oral microbiomes, while most of the other skin lesion

microbiomes also showed seal-associated bacterial species, including potential pathogens. In conclusion, this study shows that bacterial transmission from grey seals to harbour porpoises by biting is highly likely and that seal oral cavities harbour many bacterial pathogens with zoonotic potential.

# 1. Introduction

The Dutch coastal regions of the North Sea are inhabited by harbour porpoises (*Phocoena phocoena*), grey seals (*Halichoerus grypus*) and common seals (*Phoca vitulina*). Here, these species form an important part of the marine ecosystem as apex predators. In the second half of the twentieth century, the numbers of all three species in the southern North Sea were low due to various factors, such as hunting, pollution, disease and reduced food availability [1–3]. However, over the last three decades, populations of all three species have increased, which has been attributed to efficient species protection, reduced pollution, recovery from disease outbreaks (e.g. phocine distemper virus), but also to migration from other regions, potentially due to a shift in prey [2–4]. Increased numbers consequently also led to increased inter-species interactions. Over the past decade, hundreds of stranded harbour porpoises with severe mutilations have been reported [5]. Recently, these mutilations have been attributed to grey seals [6,7], a species which is also known to predate on common seals and juveniles of its own kind [8–10]. Post-mortem investigations indicated that although many harbour porpoises are killed directly, some are able to escape, of which some succumb later due to infected wounds [5,11]. Bacterial species transferred from the seal oral cavity may be involved in these infections and eventual death of these animals.

Transfer of bacterial pathogens by biting is not uncommon. A well-known example among people interacting with seals is the 'seal finger' (also known as sealer's finger or spekk finger), in which a seal bite to the hands becomes infected, very likely by *Mycoplasma* [12–15], although species like *Bisgaardia* have been indicated as well [16]. Also, a genetically distinct variant of *Neisseria animaloris* was isolated from skin abscesses, lungs and other organs of several stranded harbour porpoises with traumatic injury inflicted by grey seals [17]. As *N. animaloris* has been recovered mostly from human wounds as a result of cat or dog bites [18], a similar mode of transmission from seals to harbour porpoises was suspected. Transmission among seals by biting was also suspected for *Campylobacter pinnipediorum*, which has been detected in both skin abscesses and oral cavities of seals [19].

It has been shown previously that marine mammal species display unique microbiomes, which are distinct from the seawater microbiomes [20]. Microbiome composition differed between seals and dolphins, and between the various body parts examined. As such, the lesions observed in harbour porpoises may contain distinct signatures of the seal oral microbiome. The aim of the study is threefold: firstly, to identify the bacterial diversity and inter-species differences in the oral cavities of grey seals, common seals and harbour porpoises; secondly, to assess commonality in the bacterial diversity found in grey seal oral cavities and bite wounds (both acute and chronic) on harbour porpoises to assess the potential transfer of bacteria and thirdly, to provide insights in the bacterial species with zoonotic potential present in the oral cavities of these three marine mammals, using a 16S rRNA-based microbiome sequencing approach.

# 2. Material and methods

## 2.1. Study population and sample collection

Samples were collected from 21 stranded harbour porpoises (*Phocoena phocoena*), of which 20 had skin lesions ascribed to seal attacks, nine grey seals (*Halichoerus grypus*) and eight common seals (*Phoca vitulina*) from various coastal sites in The Netherlands (table 1). Eight harbour porpoises were probably directly killed by seal attack and did not show signs of wound healing or infection of the skin lesions (acute bite wounds; cases 2013–2014, table 1); 12 animals showed extensive infection of the skin lesions (chronic bite wounds; cases 2016–2018, table 1). All harbour porpoises were in a very fresh to fresh condition at the time of necropsy and tissue sampling, with time between death and tissue sampling ranging from a few hours to a few days. Live wild grey seals (7) and common seals (5) were sampled upon admission to a seal rehabilitation centre. Five other seals were sampled during post-mortem examination, which was conducted approximately two days after death on two seals, while the other three were temporarily frozen prior to examination. All seals were juvenile animals

**Table 1.** Case descriptions and number of samples included in this study. Hg, *Halichoerus grypus*, grey seal; Pv, *Phoca vitulina*, common seal; Pp, *Phocoena phocoena*, harbour porpoise. M, male; F, female.

| case ID | species | age class | sex | condition | stranding location | stranding date | cause of death | description of lesions | oral | skin | lesion | remarks |
|---|---|---|---|---|---|---|---|---|---|---|---|---|
| 31601110402 | Hg | juvenile | M | dead | Friesland | 11-1-2016 | emaciation | chronic, severe, necrotic bite trauma head with histologic inflammatory infiltrates, haemorrhage and accumulation of bacteria | 1 | 0 | 0 | |
| 31601293502 | Hg | juvenile | M | dead | Friesland | 28-1-2016 | infectious disease | n/a | 1 | 0 | 0 | pup (3–4 days) without teeth |
| HG16-014 | Hg | juvenile | M | alive | Noord-Holland | 8-1-2016 | n/a | n/a | 1 | 0 | 0 | |
| HG16-025 | Hg | juvenile | M | alive | Noord-Holland | 10-1-2016 | n/a | n/a | 1 | 0 | 0 | |
| HG16-027 | Hg | juvenile | M | alive | Friesland | 11-1-2016 | n/a | n/a | 1 | 0 | 0 | |
| HG16-043 | Hg | juvenile | F | alive | Noord-Holland | 19-1-2016 | n/a | n/a | 1 | 0 | 0 | |
| HG16-062 | Hg | juvenile | F | alive | Friesland | 24-1-2016 | n/a | n/a | 1 | 0 | 0 | |
| HG16-069 | Hg | juvenile | M | alive | Friesland | 28-1-2016 | n/a | n/a | 1 | 0 | 0 | |
| HG16-070 | Hg | juvenile | F | alive | Friesland | 28-1-2016 | n/a | n/a | 1 | 0 | 0 | |
| 316032103902 | Pv | juvenile | M | dead | Zuid-Holland/Zeeland | Unknown | unknown | n/a | 1 | 0 | 0 | |
| 31603210402 | Pv | juvenile | M | dead | Zuid-Holland/Zeeland | Unknown | unknown | n/a | 1 | 0 | 0 | |
| 31603210410 | Pv | juvenile | F | dead | Zuid-Holland | 2-1-2016 | unknown | n/a | 1 | 0 | 0 | |
| PV16-013 | Pv | juvenile | M | alive | Friesland | 7-1-2016 | n/a | n/a | 1 | 0 | 0 | |
| PV16-015 | Pv | juvenile | M | alive | Noord-Holland | 8-1-2016 | n/a | n/a | 1 | 0 | 0 | |
| PV16-018 | Pv | juvenile | F | alive | Friesland | 8-1-2016 | n/a | n/a | 1 | 0 | 0 | |
| PV16-026 | Pv | juvenile | M | alive | Noord-Holland | 10-1-2016 | n/a | n/a | 1 | 0 | 0 | |
| PV16-029 | Pv | juvenile | M | alive | Friesland | 11-1-2016 | n/a | n/a | 1 | 0 | 0 | |
| UT1004 | Pp | juvenile | M | dead | Zeeland | 20-8-2013 | grey seal attack | acute sharp edged mutilation throat with associated bite lesions. Acute bilateral tailstock lesion with histologic bacteria and haemorrhage | 0 | 0 | 1 | case tested positive for grey seal DNA (van Bleijswijk *et al.* [6]) |

(*Continued.*)

**Table 1.** (*Continued.*)

| case ID | species | age class | sex | condition | stranding location | stranding date | cause of death | description of lesions | number of samples | | | remarks |
| --- | --- | --- | --- | --- | --- | --- | --- | --- | --- | --- | --- | --- |
| | | | | | | | | | oral | skin | lesion | |
| UT1007 | Pp | adult | F | dead | Noord-Holland | 1-10-2013 | grey seal attack | acute large sharp edged mutilation torso. Multifocal acute bite lesions on multiple locations with histologic haemorrhage | 0 | 0 | 1 | case tested positive for grey seal DNA (van Bleijswijk et al. [6]) |
| UT1020 | Pp | juvenile | F | dead | Noord-Holland | 19-11-2013 | grey seal attack | acute large sharp edged mutilation torso. Multifocal acute bite lesions on multiple locations with histologic haemorrhage | 0 | 0 | 1 | |
| UT1292 | Pp | juvenile | F | dead | Noord-Holland | 15-12-2013 | grey seal attack | acute sharp edged mutilation throat. Multifocal acute bite lesions on multiple locations that histologically present haemorrhage | 0 | 1 | 2 | case tested positive for grey seal DNA (van Bleijswijk et al. [6]) |
| UT1300 | Pp | adult | M | dead | Zuid-Holland | 11-1-2014 | grey seal attack | acute sharp edged mutilation throat with associated bite lesions with histologic haemorrhage | 0 | 1 | 3 | |
| UT1305 | Pp | adult | M | dead | Zuid-Holland | 23-12-2013 | grey seal attack | acute sharp edged mutilation throat with associated bite lesions with histologic no haemorrhage observed | 0 | 0 | 4 | |
| UT1311 | Pp | adult | F | dead | Zeeland | 3-3-2014 | grey seal attack | acute large sharp edged mutilation torso. Multifocal acute bite lesions on multiple locations with histologic haemorrhage | 0 | 1 | 3 | |
| UT1312 | Pp | juvenile | F | dead | Zeeland | 5-3-2014 | grey seal attack | acute large sharp edged mutilation torso. Multifocal acute bite lesions on multiple locations with histologic haemorrhage | 0 | 1 | 2 | |
| UT1495 | Pp | juvenile | M | dead | Zuid-Holland | 17-2-2016 | emaciation | chronic, severe, necropurulent lesion on fluke with histologic inflammatory infiltrates, haemorrhage and accumulation of bacteria | 1 | 0 | 1 | |
| UT1503 | Pp | juvenile | F | dead | Noord-Holland | 3-3-2016 | grey seal attack | acute large sharp edged mutilation torso. Multifocal acute bite lesions on multiple locations with histologic some haemorrhage and accumulation of bacteria | 0 | 0 | 1 | |
| UT1505 | Pp | juvenile | F | dead | Zeeland | 5-3-2016 | grey seal attack | acute large sharp edged mutilation torso. Multifocal acute bite lesions on multiple locations with histologic some vacuolization | 1 | 0 | 1 | |
| UT1506 | Pp | juvenile | F | dead | Zeeland | 5-3-2016 | grey seal attack | acute large sharp edged mutilation torso. Multifocal acute bite lesions on multiple locations with histologic haemorrhage | 0 | 0 | 1 | |

(*Continued.*)

**Table 1.** (*Continued.*)

| case ID | species | age class | sex | condition | standing location | stranding date | cause of death | description of lesions | number of samples | | | remarks |
|---|---|---|---|---|---|---|---|---|---|---|---|---|
| | | | | | | | | | oral | skin | lesion | |
| UT1509 | Pp | juvenile | F | dead | Zeeland | 11-3-2016 | grey seal attack | acute large sharp edged mutilation torso. Multifocal acute bite lesions on multiple locations with histologic haemorrhage and accumulation of bacteria | 1 | 0 | 1 | |
| UT1513 | Pp | juvenile | F | dead | Zeeland | 22-3-2016 | infectious disease | multifocal scars on multiple location and acute bite lesions tailstock and fluke. Histologic inflammatory infiltrates, haemorrhage and accumulation of bacteria | 1 | 0 | 1 | |
| UT1514 | Pp | juvenile | F | dead | Zeeland | 22-3-2016 | infectious disease | chronic, severe, necropurulent, bilateral tailstock lesion with histologic inflammatory infiltrates, haemorrhage and accumulation of bacteria | 1 | 1 | 3 | |
| UT1535 | Pp | adult | F | dead | Zeeland | 28-7-2016 | infectious disease | none | 1 | 0 | 0 | |
| UT1610 | Pp | adult | M | dead | Noord-Holland | 26-7-2017 | infectious disease | almost completely healed bilateral tailstock lesion (scar) with histologic inflammatory infiltrates and haemorrhage | 0 | 0 | 1 | |
| UT1635 | Pp | adult | M | dead | Zuid-Holland | 12-12-2017 | infectious disease | chronic, severe, necropurulent, lesion throat with histologic inflammatory infiltrates and haemorrhage. Almost completely healed bilateral tailstock lesion (scar) with histological fibrosing, necrosis and accumulation of bacteria | 0 | 0 | 1 | |
| UT1648 | Pp | juvenile | M | dead | Zeeland | 14-2-2018 | infectious disease | almost completely healed bilateral tailstock lesion (scar) with histological fibrosing, some haemorrhage, accumulation of bacteria and necrosis | 0 | 0 | 1 | |
| UT1656 | Pp | adult | M | dead | Noord-Holland | 18-3-2018 | grey seal attack | chronic, severe, necropurulent, bilateral tailstock lesion with histologically fibrosing, accumulation of bacteria and necrosis. acute large sharp edged mutilation torso with multifocal bite lesions on multiple locations with histologic haemorrhage | 0 | 0 | 1 | |
| UT1662 | Pp | juvenile | F | dead | Noord-Holland | 25-4-2018 | infectious disease | chronic, severe, necropurulent, bilateral tailstock lesion with histologic inflammatory infiltrates, fibrosing, haemorrhage and accumulation of bacteria | 0 | 0 | 1 | |

(estimated age 3 days up to 7 months), whereas the harbour porpoises included both juvenile and adult animals. Skin lesions of three of these harbour porpoises had tested positive for the presence of grey seal DNA in a previous study [6].

Oral swab samples were collected from the tooth base of all three species (harbour porpoise $n = 6$, grey seal $n = 9$ and common seal $n = 8$). Teeth had not erupted on one juvenile seal (HG16-014) and the gums were sampled. Additionally, swab samples were collected from skin lesions ($n = 31$), with a preference for deep puncture wounds, and unaffected skin ($n = 5$) of harbour porpoises, resulting in a total of 59 samples (table 1). The unaffected skin samples were taken from the surface of intact skin of the mutilated harbour porpoises and included as controls for the skin lesion samples. Swab samples were stored frozen at −20°C until DNA extraction.

## 2.2. DNA extraction, library preparation and 16S rRNA gene sequencing

Swab samples were extracted in 1 ml FE buffer (150 mM NaCl, 1 mM EDTA). Of the suspension, 200 µl was used as input for DNA extraction using the DNeasy Blood & Tissue kit (Qiagen, Venlo, The Netherlands).

The variable V3 and V4 regions of the 16S rRNA gene were amplified and libraries were prepared following the 16S Metagenomic Sequencing Library Preparation protocol (Illumina). Next, each library was normalized, pooled and loaded onto the Illumina MiSeq platform for paired-end sequencing using the 600 cycles MiSeq Reagent Kit V3 (Illumina) generating $2 \times 300$ basepair paired-end reads.

## 2.3. Microbiome analysis

Reads of the V3 and V4 regions of the 16S rRNA gene were processed using DADA2 and the Phyloseq package [21,22] as described in the DADA2 tutorial v. 1.6 (https://benjjneb.github.io/dada2/tutorial_1_6.html). α- and β-diversity was determined using Shannon, Simpson and unweighted Unifrac [23], respectively. The complete set of R commands applied to the data is available as electronic supplementary material.

## 2.4. Phylogenetic analyses

Alignment of 16S rRNA sequences and dendrogram construction were performed using MEGA v. 6.05 [24]. A neighbour-joining dendrogram containing all *Campylobacter*, *Mycoplasma* and *Neisseria* operational taxonomic units (OTUs) was constructed, with reference taxa extracted from GenBank and bootstrap values based on 500 repetitions.

# 3. Results

## 3.1. Microbiome diversity

In total, 50 phyla were recognized (figure 1), of which 29 were considered rare (less than 10 OTUs; grouped as 'other phyla' in figure 1). Proteobacteria, Bacteroidetes, Fusobacteria and Firmicutes were the dominant phyla. Phyla diversity was higher in the skin lesion and skin microbiomes, compared with the oral microbiomes. Fusobacteria were well represented in the oral microbiomes of all host species, but less in the intact skin and skin lesion microbiomes. Nevertheless, Fusobacteria diversity was higher in the skin lesion microbiomes than in the intact skin microbiomes. One grey seal (316011100402) showed a markedly deviant microbiome composition, with Tenericutes dominating the oral microbiome, which could largely be attributed to one particular *Mycoplasma* OTU (OTU_0008), which was most closely related to *Mycoplasma equigenitalium* (95.4% sequence identity).

Total diversity for all combined samples was 9915 OTUs (electronic supplementary material, table S1), with an average of 578 OTUs per sample. Average oral diversity per sample for harbour porpoise, grey seal and common seal was 607, 444 and 417 OTUs, respectively. Harbour porpoise skin lesions showed highest average bacterial diversity per sample (678 OTUs), which was considerably higher than the average diversity of the skin samples (425 OTUs).

An OTU with identical 16S rRNA sequence to *Bisgaardia* genomospecies 1 strain M2461/98/1 isolated from seals [25] showed highest read counts and was most widespread among all samples, being the only OTU present in all samples from seals and harbour porpoises.

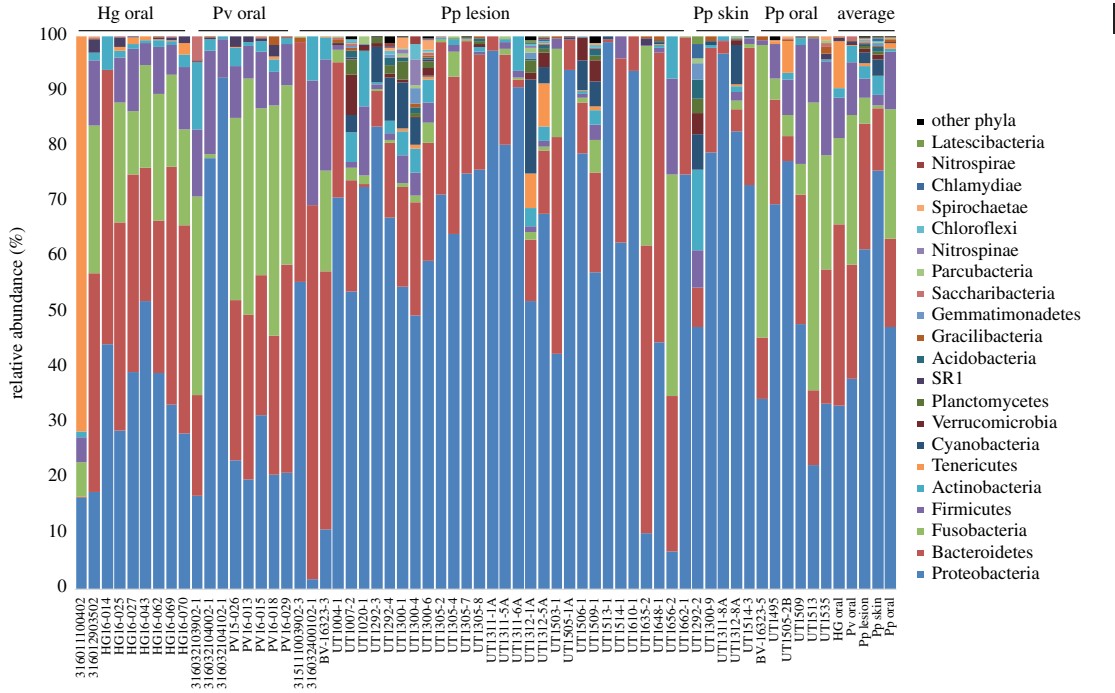

**Figure 1.** Distribution of phyla for all microbiomes, clustered per sample type, including the average distribution of phyla for each sample type. Hg, *Halichoerus grypus*, grey seal; Pv, *Phoca vitulina*, common seal; Pp, *Phocoena phocoena*, harbour porpoise.

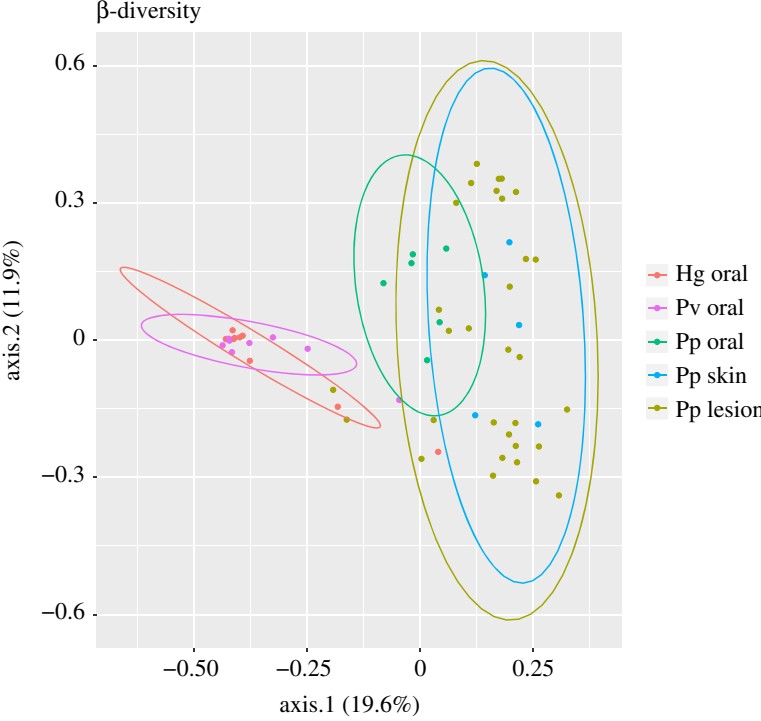

**Figure 2.** PCOA plot showing the β-diversity analysis based on unweighted Unifrac for the microbiomes of all samples included in this study. Each point represents a sample/microbiome. Circles represent the 95% confidence interval for each sample type. Hg, *Halichoerus grypus*, grey seal; Pv, *Phoca vitulina*, common seal; Pp, *Phocoena phocoena*, harbour porpoise.

## 3.2. Seal and harbour porpoise oral microbiome diversity

The oral microbiome diversity of both seal species was highly similar, but clearly deviant from the harbour porpoise oral microbiome diversity (figure 2; electronic supplementary material, figure S1). Also, average oral diversity per sample was markedly higher for harbour porpoises (607 OTUs), than for grey seals (444 OTUs) and common seals (417 OTUs).

In the oral cavities of both seal species, multiple *Bisgaardia*, *Fusobacterium*, *Oceanivirga* and *Porphyromonas* OTUs were among the most abundant OTUs based on read counts (electronic supplementary material, table S1). Other highly abundant OTUs were *Neisseria zalophi* (OTU_0003), *Bergeyella* (OTU_0007), *Mycoplasma* (OTU_0008), *Streptobacillus* (OTU_0006), *Psychrobacter* (OTU_0014), *Ornithobacterium* (OTU_0019), *Marinifilum* (OTU_0023) and *Campylobacter pinnipediorum* (OTU_0035). In addition, *Streptococcus phocae* (OTU_0021) and *Arcanobacterium phocisimile* (OTU_0038) were present in all seal oral microbiomes.

Based on read counts, the harbour porpoise oral cavity was dominated by various *Phocoenobacter* OTUs, a genus which is currently represented by one species, *P. uteri* [26]. Furthermore, multiple *Fusobacterium*, *Porphyromonas*, *Psychrobacter* and *Fusibacter* OTUs showed high abundance. An *Arcobacter* OTU (OTU_0073), most closely related to *A. aquimarinus* and *A. butzleri*, was the sixth most abundant OTU. Two *Helicobacter* OTUs (OTU_0267 and OTU_0204) most closely related to *H. cetorum* (98% homology) also showed high read counts in the harbour porpoise oral cavities, but were not present in seal oral cavities.

Compared with the seal oral cavity, potential pathogenic bacterial species and genera appeared to be less abundant in the harbour porpoise oral cavity.

## 3.3. Seal oral bacteria in harbour porpoise lesions

β-diversity was clearly distinct between seals and harbour porpoise microbiomes (figure 2; electronic supplementary material, figure S1). Interestingly, both intra- and inter-species β-diversity was highly similar for most of the seal oral microbiomes. β-diversity showed more variation for the different harbour porpoise microbiomes, nevertheless the microbiomes of all harbour porpoise sample types showed overlap, particularly the skin and skin lesion microbiomes. β-diversity of skin lesions from the same animal was highly divergent in most cases. β-diversities of two skin lesion microbiomes (UT1656-2 and BV-16323-3 (UT1514)) were similar to those of the grey seal oral microbiomes, indicating the presence of seal oral bacteria and supporting transfer of bacteria from the grey seal oral cavity to the harbour porpoise lesions. Additionally, two seal oral microbiomes were highly similar to those of two skin lesions (UT1635-2 and 316032400102-1 (UT1514)). These were within the 95% confidence interval of the skin lesion microbiomes, but outside the 95% confidence interval of the harbour porpoise skin and oral microbiomes. All four skin lesion microbiomes which showed similar β-diversity to seal oral microbiomes were from infected skin lesions of harbour porpoises which initially escaped from grey seal attack and not from skin lesions of animals directly killed by attack, including those skin lesions in which grey seal DNA was detected. The β-diversity of HG16-014, a grey seal pup without teeth, was clearly distinct from the β-diversity of the other seals.

Although bacterial transfer from seals to harbour porpoise lesions was apparent in 12.9% (4/31) of the lesion samples based on β-diversity analyses, corresponding to 14.3% (3/21) of the included porpoises, most other lesion microbiomes also included bacteria which most likely originated from the seal oral cavity.

The OTU with the highest read counts in the harbour porpoise skin lesions belonged to the *Porphyromonas* genus (OTU_0009). This OTU also showed high read counts in grey seal oral cavities, while being rare or absent in other sample types. *Porphyromonas* species are mostly anaerobic, and typically associated with the oral cavity, but also with infections in various regions of the body [27].

*Streptococcus phocae* (OTU_0021) showed high read counts in harbour porpoise skin lesions and seal oral cavities, in particular in grey seals. It occurred in all seal oral cavities and part of the harbour porpoise lesions (9/31) with high read counts, while being absent or present with lower read counts in other sample types.

*Arcanobacterium phocisimile* (OTU_0038) occurred in all seal oral cavities and in harbour porpoise skin lesions (4/31) with high read counts, while being scarce in other sample types. This OTU was present in three out of four skin lesion microbiomes which resembled seal oral microbiomes based on β-diversity analysis. This species has been isolated from both apparently healthy and diseased common seals and its pathogenic importance is unclear [28].

A *Streptobacillus* OTU (OTU_0006) showed high read counts in the oral cavities of both seal species (16/17) and in one harbour porpoise skin lesion (UT1656-2), while being absent or present with low read counts in other sample types.

The 30 *Mycoplasma* OTUs were often mutually exclusive, i.e. either associated with seal or harbour porpoise. However, four *Mycoplasma* OTUs which were most abundant in both seal species also occurred in harbour porpoise skin lesions, while being less abundant or absent in other harbour porpoise sample types. The most abundant *Mycoplasma* OTU (OTU_0008) in both seal species (11/17),

although with higher abundance in grey seals, was also detected in seven harbour porpoise skin lesions with low to moderate read counts. *Mycoplasma phocicerebrale* (OTU_0338) was widespread (8/17) in both seal species with moderate to high read counts. It was also found in one skin lesion (BV-16323-3 (UT1514)) with moderate read counts, but not in other harbour porpoise samples.

*Neisseria zalophi* (OTU_0003) occurred in all seal oral cavities with high read counts. It was widespread (18/31) in harbour porpoise skin lesions with low to moderate read counts, with high read counts in one sample (UT1312-5A).

A *Fusobacterium* OTU (OTU_0011) showed high read counts in 94.1% (16/17) oral cavities of both seal species and in one harbour porpoise skin lesion (BV-16323-3 (UT1514)), while being rare or absent in other sample types.

Diversity of *Bergeyella* OTUs was high in both seals and harbour porpoises and mostly showed a distinct host association with either seal or harbour porpoise. However, the most abundant *Bergeyella* OTU in seals (OTU_0007), both in read count and prevalence (17/17), was also present in harbour porpoise skin lesions (21/31) with moderate read counts.

## 3.4. Potential pathogens in seal and harbour porpoise microbiomes

A total of 30 different *Mycoplasma* OTUs was present in all sample types. Six *Mycoplasma* OTUs were found exclusively in both seal species, 17 OTUs were found exclusively in harbour porpoises, and seven were found in both seals and harbour porpoises. Phylogenetic analysis showed a high diversity, including many potential novel species (electronic supplementary material, figure S2). Multiple distinct clades were recognized, which were associated with either seal or harbour porpoise. Twelve OTUs formed a large distinct clade which included *M. equigenitalium* and *M. elephantis*, with sub-structuring in two clades associated with either seal or harbour porpoise. *Mycoplasma phocicerebrale* (OTU_0338) was present in both seal species, as was *M. phocirhinis* (OTU_6604), albeit with low read counts in four samples. One *Mycoplasma* OTU (OTU_0008) showed highest overall read count in grey seal oral cavities, although this was mainly attributed to one sample (316011100402).

Nine *Neisseria* OTUs were detected in total. Four OTUs were detected exclusively in both seal species, two were detected exclusively in harbour porpoises and three in both seals and harbour porpoises. All were closely related to previously described species (figure 3). Five OTUs were genetic variants of *N. zalophi*. *Neisseria animaloris* (OTU_0689) was present in grey seal oral cavities (4/9) with low to moderate read counts, but absent from other sample types. *Neisseria zalophi* (OTU_0003) was widespread (17/17) with high read counts in both seal species (third highest read count) and was also present in harbour porpoises with low read counts, although read counts were higher in skin lesions, with skin lesion UT1312-5A showing highest read counts.

*Campylobacter* diversity totalled 22 OTUs. Seven OTUs were detected exclusively in both seal species, seven exclusively in harbour porpoises, and eight in both seals and harbour porpoises. Eight OTUs formed a clade with the recently described *C. blaseri*, and which may comprise multiple novel species (figure 4). A distinct clade most closely related to *C. rectus* and *C. showae* contained four OTUs which probably represent novel species. Three OTUs formed a sister group to the *Campylobacter* genus, which potentially represents a novel genus most closely related to *Campylobacter*. *Campylobacter pinnipediorum* (OTU_0035) was widespread in both seal species (15/17) and showed high read counts in the oral cavities of both seal species, particularly in grey seal (tenth highest read count), while being the second most abundant OTU in one grey seal sample (316011100402). However, read counts were low in harbour porpoise skin lesions (1–15 reads in 6/31 samples). *Campylobacter* OTU_0223, most closely related to *C. rectus* and *C. showae*, was well represented in most seal oral cavities (14/17) and in four skin lesions, of which two showed moderate to high read counts (UT1656-2 and BV-16323-3 (UT1514)), while being absent from other samples. Notably, these two skin lesion microbiomes also showed most similar β-diversity to the seal microbiomes.

In addition to the aforementioned potential pathogens, a *Brucella* OTU (OTU_4728) was detected in the oral cavities of both seal species (5/17) and one harbour porpoise, and in one harbour porpoise skin lesion, but all with low read counts (1–21).

## 4. Discussion

Despite living in the same aquatic environment, seal and harbour porpoise microbiomes are clearly distinct, as has been shown for other sympatric pinniped and cetacean species [20]. Interestingly, based on β-diversity analysis using unweighted Unifrac, four of the 31 harbour porpoise skin lesion

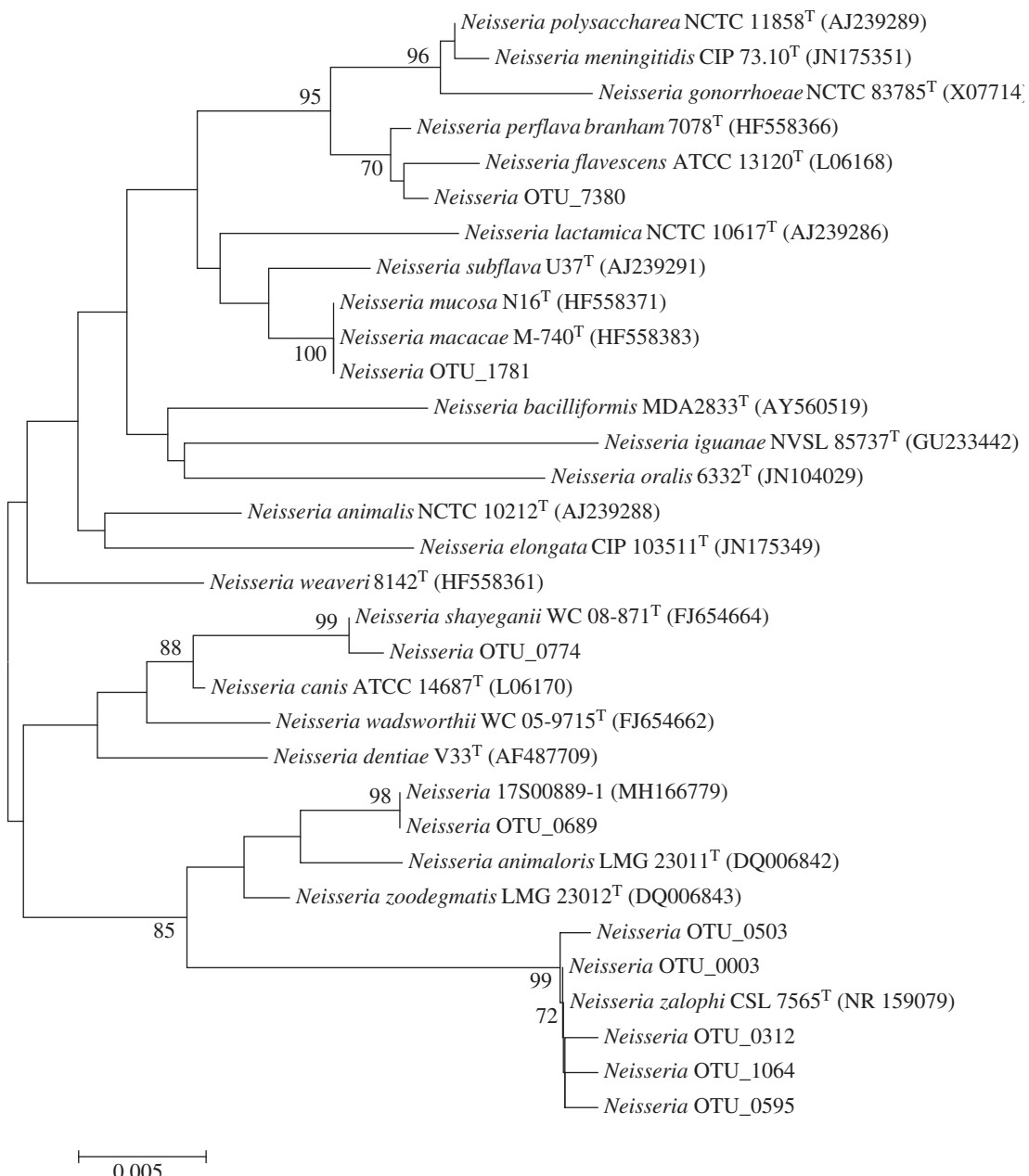

**Figure 3.** Phylogeny (neighbour-joining, 500 bootstraps) based on partial 16S rRNA sequences of all *Neisseria* OTUs in this study and reference species.

microbiomes were highly similar to the seal oral microbiomes, consistent with bacterial transfer. This was supported by the identification of seal-associated OTUs in these skin lesions, such as specific *Campylobacter*, *Fusobacterium*, *Mycoplasma* and *Streptobacillus* OTUs. Bacterial transfer from seals to harbour porpoises by biting is highly likely based on these results: in addition to the four harbour porpoise skin lesion microbiomes that were similar to seal oral microbiomes, many other skin lesion microbiomes contained OTUs typically associated with seals or the oral cavity in general. In this respect, *Mycoplasma* may be a good indicator of bacterial transfer, as these species are often highly associated with a particular host. The association of distinct *Mycoplasma* clades with either seals or harbour porpoises supported a high level of host adaptation. Notably, the most abundant *Mycoplasma* OTUs from seals were also detected in harbour porpoise skin lesions, while being less abundant or absent in other harbour porpoise sample types. This also included *M. phocicerebrale*, a species associated with the seal oral cavity [29], which was detected in the oral cavities of both seal species, although more prevalent in grey seal, and in a harbour porpoise skin lesion.

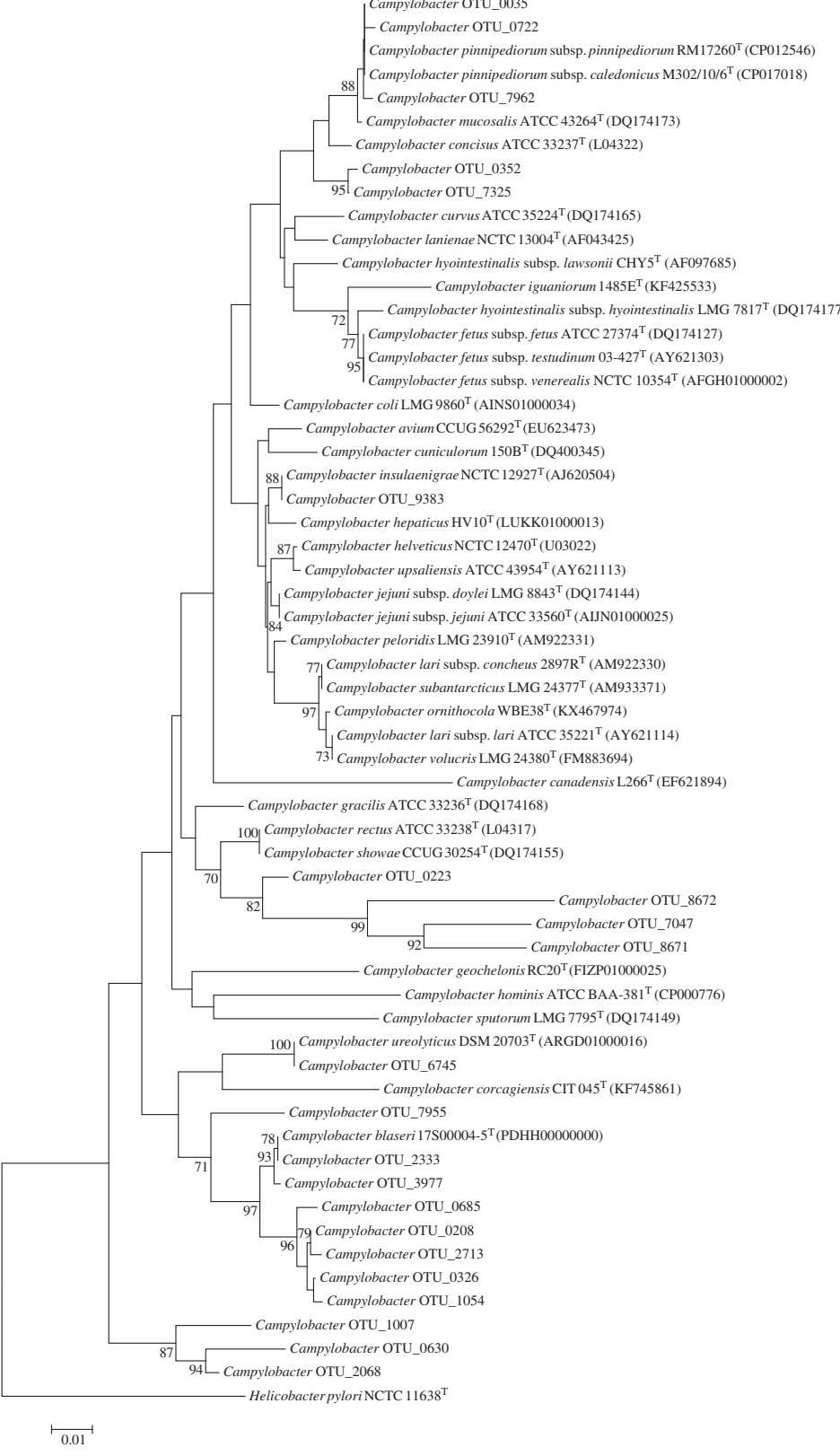

**Figure 4.** Phylogeny (neighbour-joining, 500 bootstraps) based on partial 16S rRNA sequences of all *Campylobacter* OTUs in this study and reference species.

The oral microbiome of both seal species was highly similar, which makes it hard to attribute the seal-associated bacteria in the harbour porpoise skin lesions with high certainty to a particular species. Nevertheless, as grey seal DNA has been detected in part of the skin lesions, these bacteria probably originated from this seal species.

Harbour porpoise skin lesion microbiomes were highly variable, even among samples from the same individual animal. Although OTUs typically associated with the seal oral cavity were present in most harbour porpoise skin lesions, the composition was mostly highly variable and did not directly reflect the seal oral microbiome composition. Nevertheless, OTUs that were highly abundant in seal oral cavities were often also detected in harbour porpoise skin lesions. The apparent absence of a clear seal oral bacterial signature, even in skin lesions which were positive for grey seal DNA, and seemingly randomly distributed infectious agents from the seal oral cavity can be explained by several factors. While overall similar, seals have slightly different oral microbiomes at an individual level, leading to various bacterial transmission patterns. A small part of the seal oral bacterial diversity may be transmitted, compared with the diversity already present on the harbour porpoise skin and skin lesions. Indeed, β-diversity of the unaffected skin and skin lesions was highly congruent, indicating that unaffected skin lesions share a similar bacterial diversity. Not all seal oral bacteria will be adapted to survive in the lesion, which are highly divergent niches. Although deeper puncture wounds were selected, part of the transmitted bacteria may have been flushed or cleared from the lesion. Also, post-mortem changes cannot be excluded. The oral microbiomes of dead seals were divergent in some cases. It is unclear whether the divergent microbiomes of the dead animals reflects the microbiome of the animals while alive or whether it can be ascribed to bacterial changes after death.

Skin lesion microbiomes showed highest average diversity per sample, and diversity was notably higher than diversity of the unaffected skin microbiomes. This high diversity may be explained by a higher number of skin lesion samples compared with the other sample types. However, most likely this may be attributed to added bacterial diversity from the seal oral cavity, and from water and sediment upon stranding, which may accumulate in the lesions. Many species and genera uniquely present in skin lesions were associated with seawater and marine sediment, indicating contamination or opportunistic colonization of the skin lesions from these sources.

Notably, the oral microbiomes of both seal species were highly similar, despite collection from different locations, without having contact with each other prior to sampling. All seals included in this study were juvenile animals, while the harbour porpoise lesions are probably ascribed to adult grey seals. The microbiomes of juvenile and adult seals may differ, as the microbiome develops and diverges while ageing [30]. Nevertheless, the presence of many bacterial species which previously have been associated with adult seals suggests that many components of the adult oral microbiome were already present in the juvenile oral microbiome. However, the oral microbiome of one very young juvenile grey seal, which did not have teeth yet at the moment of sampling, was divergent, probably due to sampling of the gums instead of the tooth base.

Bacteria detected in the seal oral cavity and in harbour porpoise lesions included bacteria which have zoonotic potential. Some known and potential zoonotic agents from the seal oral cavity were *Bergeyella* sp., *Brucella* spp., *Campylobacter* spp., including *C. pinnipediorum*, *Fusobacterium* spp., *Mycoplasma* spp., including *M. phocicerebrale*, *Neisseria animaloris*, *Streptobacillus* spp. and *Streptococcus phocae*. Compared with the seal oral cavity, few bacterial species or genera typically associated with disease appeared to dominate the harbour porpoise oral cavity. However, this may also be attributed to the lower frequency of human interactions with, and biting incidents caused by harbour porpoises, compared with seals, thereby underestimating the pathogenic potential of the harbour porpoise oral microbiota.

The most abundant *Bergeyella* OTU in seals (OTU_0007) was most closely related to *Bergeyella zoohelcum* (94.2% sequence identity), which is considered an uncommon zoonotic pathogen typically associated with cat or dog bites [31].

*Brucella* was identified at low densities in harbour porpoises and the oral cavities of both seal species. *Brucella* has been identified in harbour porpoises previously, predominantly in the lungs [32,33]. It has previously been detected in multiple organs of both grey and common seals [34]. Based on 16S rRNA alone, many *Brucella* species cannot be differentiated. Nevertheless, *Brucella* species often have distinct host specificity, with *B. ceti* occurring in harbour porpoises and other cetaceans and *B. pinnipedialis* in pinnipeds [35,36]. *Brucella* can be highly infectious and marine species are known to infect humans [37]. Although the zoonotic potential of marine *Brucella* species is considered low [38], infections in humans can be severe, and the presence of *Brucella* in seal oral cavities may facilitate transmission to humans by biting.

Although *C. pinnipediorum* was isolated from pinniped abscesses previously [19] and was very abundant in seal oral cavities, it was scarce in harbour porpoise lesions. This suggests that although this *Campylobacter* species may be transferred by biting, it is not a significant infectious agent in the harbour porpoise skin lesions analysed in this study. Nevertheless, given the abundance in seal oral cavities and seal skin abscesses, zoonotic potential cannot be excluded. The same may hold true for

*Campylobacter* OTU_0223, which was identified in seal oral cavities and in two harbour porpoise skin lesions which showed most similar β-diversity to the seal oral cavities.

Fusobacteria diversity was higher in the skin lesion microbiomes, compared with the intact skin microbiomes. Fusobacteria are often well represented in oral cavities and infections of soft tissue, skin and muscle, including animal and human bite wounds [39]. The finding of large numbers of *Fusobacterium*, as well as other anaerobes such as *Porphyromonas*, emphasizes the importance of anaerobe culture for diagnostics.

*Mycoplasma phocicerebrale*, *M. phocidae* and *M. phocirhinis* were isolated from the oral cavities and infected wounds of common and grey seals [29], with *M. phocicerebrale* and *M. phocidae* consistently identified from infections, while *M. phocicerebrale* has also previously been reported from harbour porpoise lungs [40]. *Mycoplasma* is often identified as the cause of infection after a seal bite in humans [13–15]. In this study, a large variety of *Mycoplasma* OTUs, including *M. phocicerebrale* and *M. phocirhinis*, were identified in both seals and harbour porpoises. Many of these probably represent novel species, which may include species which can be pathogenic to humans.

A seal-associated genetic variant of *Neisseria animaloris* was present in four grey seal oral cavities and absent from other sample types, including harbour porpoise lesions. Nevertheless, at present, this *N. animaloris* variant is only known from the grey seal oral cavity and from internal organs and skin lesions of harbour porpoises attacked by grey seals [17], suggesting that *N. animaloris* transfer from grey seal to harbour porpoise and subsequent infection is plausible. The apparent absence of this *N. animaloris* variant in harbour porpoise skin lesions in the present study could be explained, as the infected porpoises from the previous study [17] were not included, and not all grey seals appear to carry *N. animaloris*. *Neisseria animaloris* has been recovered from human wounds as a result of cat or dog bites [18] and the *N. animaloris* variant from seals may have similar zoonotic potential.

*Streptobacillus* OTU (OTU_0006) was closely related to *S. notomytis* and *S. moniliformis*, which have both been implicated in rat-bite fever, a systemic infection caused by rat bites [41,42]. In humans, *S. moniliformis* infection has a mortality rate of 13% when untreated [41].

*Streptococcus phocae* is a facultative anaerobic species which has previously been isolated from common and grey seals and has often been implicated in the final cause of death of seals infected with phocine distemper virus [43]. *Streptococcus phocae* has been isolated from other pinniped species, harbour porpoises and sea otters (*Enhydra lutri*). Pathologic manifestations of *S. phocae*-associated disease included localized, as well as systemic, inflammatory lesions [44].

Bacterial species and genera typically associated with disease appeared to be less abundant in the harbour porpoise oral cavity, compared with the seal oral cavity. However, a notable potential pathogen solely present in harbour porpoise samples was a genetic variant of *Helicobacter cetorum*, which has been implicated in gastritis in cetaceans previously [45]. Interestingly, it was highly abundant in harbour porpoise oral cavities, but also consistently detected in multiple unaffected skin and skin lesion samples from two individual porpoises, which may indicate bacterial contamination from the oral cavity of the animals themselves or that the skin lesions may be inflicted by a conspecific animal or other cetacean.

The high abundance of potential pathogens in the seal oral cavity and the possibility for severe infection in humans after a seal bite make porpoise mortality due to infections caused by grey seal bites a plausible scenario [17]. In conclusion, this study shows that bacterial transmission from grey seals to harbour porpoises is highly likely and that seal oral cavities harbour many bacterial pathogens with zoonotic potential.

Ethics. The procedures conducted on living animals in the study were combined with veterinary diagnostic and therapeutic acts and were therefore not considered to cause any additional discomfort. Consequently, the study was not considered an animal experiment under the Dutch 'Animals under experiments act', making an assessment by an animal ethics committee unnecessary. Under the same act, collection of tissue from animals which died from natural causes is not regarded as an animal experiment. Sampling of live animals was performed by a veterinarian specialized in marine mammals. The seal rehabilitation centre has been granted exemption from wildlife protection laws (permit no. FF/75/2012/015). The Dutch Ministry of Agriculture, Nature and Food Quality commissioned transport of animals which died from natural causes and further post-mortem investigation and granted exemption from wildlife protection laws (Nature Act, commissioning no. Wnb/2018/039).

Data accessibility. The datasets supporting this article have been uploaded as part of the electronic supplementary material.

Authors' contributions. A.G. made substantial contributions to analysis and interpretation of data; A.R.-G. made substantial contributions to acquisition of data; A.L.Z. made substantial contributions to conception and design and to analysis and interpretation of data; B.D. made substantial contributions to conception and design and to analysis and interpretation of data; J.R. made substantial contributions to acquisition of data; J.A.W. made substantial contributions to conception and design and to analysis and interpretation of data; L.L.I. made substantial

contributions to conception and design, to acquisition of data and to analysis and interpretation of data; M.J.G. made substantial contributions to conception and design, to acquisition of data, to analysis and interpretation of data, and drafted the manuscript. All authors critically revised the manuscript, gave final approval for publication and agree to be held accountable for the work performed therein.

Competing interests. We have no competing interests.

Funding. This work received no specific grant from any funding agency.

Acknowledgements. We would like to thank all the volunteers of the Dutch stranding network who report and collect marine mammals for rehabilitation or post-mortem examination, and the veterinary pathologists, technicians, students, caretakers and volunteers involved. Specifically we would like to acknowledge Arjen Timmerman, Bennie van Heeswijk, Erwin Raangs, Jooske IJzer, Liliane Solé, Lineke Begeman, Maarten van Putten and Marja Kik.

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
