## [Reviewer comments · Royal Society Open Science]

**After the bite: bacterial transmission from grey seals
(*Halichoerus grypus*) to harbour porpoises (*Phocoena
phocoena*)**

Maarten J. Gilbert, Lonneke L. IJsseldijk, Ana Rubio-García, Andrea Gröne, Birgitta Duim,
John Rossen, Aldert L. Zomer and Jaap A. Wagenaar

Article citation details

R. Soc. open sci. **7**: 192079.

<http://dx.doi.org/10.1098/rsos.192079>

Review timeline

Original submission: 28 November 2019

1st revised submission: 20 February 2020

2nd revised submission: 12 March 2020

Final acceptance: 24 March 2020

Note: Reports are unedited and appear as
submitted by the referee. The review history
appears in chronological order.

Review History

RSOS-192079.R0 (Original submission)

Review form: Reviewer 1 (Geoffrey Foster)

Is the manuscript scientifically sound in its present form?

Yes

Are the interpretations and conclusions justified by the results?

Yes

Is the language acceptable?

Yes

Do you have any ethical concerns with this paper?

No

Have you any concerns about statistical analyses in this paper?

No

Recommendation?

Accept with minor revision (please list in comments)

Comments to the Author(s)

The authors present a microbiome study of the oral flora of seals and harbour porpoises and compare these with the flora of skin bite lesions in harbour porpoises, as well as samples from unaffected skin. Work published recently, which included some of the authors on the current submission, documented attack by grey seals on harbour porpoises and the isolation of *Neisseria animaloris* from lesions following these attacks, which ultimately resulted in the death of the animals.

The current ms provides evidence that most samples from porpoise lesions contained microbiomes with organisms likely derived from seals during bite attacks, including four samples that were highly similar to grey seal oral flora samples. Many of these organisms are noted as potentially pathogenic and zoonotic. Five samples collected from the unaffected skin of porpoises, included as controls, were less similar to seal oral flora, thereby further supporting the transfer of organisms from the grey seals to porpoises during attacks. One caveat in this study, which the authors do point out, is that the oral flora were sampled from juvenile seals, whereas attacks on porpoises are associated with adult seals.

Interestingly, the authors did not identify *N. animaloris* from any of the porpoise lesions sampled in this paper, however, it was only detected from the mouths of four grey seals and only in low or moderate numbers. The previous paper reporting its isolation from porpoise lesions did include Dutch animals, however, it is not stated whether these animals were included in this study. This needs clarified and warrants a little more discussion.

Bacterial transfer was apparent in 4/31 porpoise lesions but 3 of these appear to be from the same porpoise (UT1514). Please also present as number of porpoises. Based on other work, is it likely that a single seal would be involved in each attack – if so add this information in text?

I did not find any comment on whether there were differences based on the site of the individual lesions. Figure 2 lists lesion sites as numbers, could this be coded such that lesion site is clearer and a comment added in the text. It may also enable easier comparison between acute and chronic lesions.

Further comments, mostly typographical include:

The finding of large numbers of *Porphyromonas* as well as other anaerobes such as *Fusobacterium* emphasises the importance of anaerobe culture for diagnostic workers. Could a comment with this advice be added.

Other points mostly typographical:

General – when writing about skin samples from five controls, it would read better to specify these as unaffected skin samples. Likewise when other sample types is used when it appears that it is only unaffected skin samples that are being affected to.

90 – change ‘one juvenile was not having teeth yet’ to ‘Teeth had not erupted on one juvenile seal’

105 – typo – ‘basepare’ should be ‘basepair’

Lines 169–173 – I find the use of two ID numbers a little confusing for these key samples. This occurs elsewhere including the tables and figures, e.g. BV-16323-3 is included in Fig 2 but is absent from Table 1.

Line 191 – change ‘particular in grey seal’ to ‘in particular in grey seals’

204 – Typo - *Streptobacillus*

227-8 – provide number of lesions/ animals?

319 – this sentence on *Fusobacterium* does not follow well from the previous sentence which is on sea water and sediment.

326 – typo ‘grey greys’

325-32 – would it be appropriate to add a reference from other mammalian hosts which have compared oral microbiomes at different ages?

327 – typo – ‘aging’

334 – change – ‘not sure’ to ‘unclear’

339 – change ‘lesion’ to ‘lesions’

350 – delete first ‘be’ before ‘also’

354-355 – Brucella has been reported from porpoise skin lesions previously: Veterinary Microbiology 90:563-580

370-2 M phocicerebrale has been reported from porpoise lungs previously: Journal of Wildlife Diseases 47:206-211. Given the typical host-adapted nature of mycoplasmas and its introduction to humans in seal finger cases, this report provides a possible reason for its finding in another host.

388-92 – How feasible is it for a porpoise to lick different parts of its own skin? Is it possible that a fellow porpoise would lick the wound of an injured animal, thereby introducing further organisms to a fresh wound?

Lines 394-396 – The authors previous N. animaloris paper should be referenced here as additional support.

Review form: Reviewer 2

Is the manuscript scientifically sound in its present form?

Yes

Are the interpretations and conclusions justified by the results?

No

Is the language acceptable?

Yes

Do you have any ethical concerns with this paper?

No

Have you any concerns about statistical analyses in this paper?

No

Recommendation?

Accept with minor revision (please list in comments)

Comments to the Author(s)

Please see attached file (Appendix A).

Decision letter (RSOS-192079.R0)

14-Feb-2020

Dear Dr Gilbert,

On behalf of the Editors, I am pleased to inform you that your Manuscript RSOS-192079 entitled "After the bite: bacterial transmission from grey seals (*Halichoerus grypus*) to harbour porpoises (*Phocoena phocoena*)" has been accepted for publication in Royal Society Open Science subject to

minor revision in accordance with the referee suggestions. Please find the referees' comments at the end of this email.

The reviewers and handling editors have recommended publication, but also suggest some minor revisions to your manuscript. Therefore, I invite you to respond to the comments and revise your manuscript.

- Ethics statement

- Data accessibility

If you wish to submit your supporting data or code to Dryad (<http://datadryad.org/>), or modify your current submission to dryad, please use the following link:
<http://datadryad.org/submit?journalID=RSOS&manu=RSOS-192079>

- Competing interests

- Authors' contributions

- Acknowledgements

- Funding statement

Because the schedule for publication is very tight, it is a condition of publication that you submit the revised version of your manuscript before 23-Feb-2020. Please note that the revision deadline will expire at 00.00am on this date. If you do not think you will be able to meet this date please let me know immediately.

Please note that Royal Society Open Science charge article processing charges for all new submissions that are accepted for publication. Charges will also apply to papers transferred to

Royal Society Open Science from other Royal Society Publishing journals, as well as papers submitted as part of our collaboration with the Royal Society of Chemistry (<https://royalsocietypublishing.org/rsos/chemistry>).

If your manuscript is newly submitted and subsequently accepted for publication, you will be asked to pay the article processing charge, unless you request a waiver and this is approved by Royal Society Publishing. You can find out more about the charges at <https://royalsocietypublishing.org/rsos/charges>. Should you have any queries, please contact openscience@royalsociety.org.

on behalf of Dr Denise Greig (Associate Editor) and Kevin Padian (Subject Editor)
openscience@royalsociety.org

Associate Editor Comments to Author (Dr Denise Greig):

This is an interesting manuscript providing additional evidence around newly described predatory behavior by grey seals on harbour porpoises. I agree with the reviewer comments, particularly around sample type, sample location, acute versus chronic lesions, and samples collected from live versus dead porpoises, as well as labelling the animals and their specimens clearly and consistently in the text and tables and figures.

I have a few additional questions detailed below:

Line 86 – when you say the seals are “juveniles”, how young do you mean? Are they pre-weaned? Recently weaned? Foraging on adult prey items? (clearly the one with unerupted teeth was pre-weaned). And a follow-up question is how does an oral microbiome develop: does it develop in utero, come from adult skin during suckling, prey items?

Line 111. Thank you for supplying your R code. It would be great if there was a little more annotation of the various commands.

Line 182. Is the 12.9% (4/31)...4 out of 31 porpoises? Or porpoise lesions?

Line 201-202. Is this result visible in Figure 2? Is there a way to label it?

Line 274. Do you mean “sympatric” instead of “synoptic”?

Lines 358-361. I agree that *Brucella* is of zoonotic concern for people, but I think all the documented human infections have come from the dolphin specific species, right? It might be worth noting that since we don't know which species you detected in the seals or porpoises.

Reviewer comments to Author:

Reviewer: 1

Comments to the Author(s)

The authors present a microbiome study of the oral flora of seals and harbour porpoises and compare these with the flora of skin bite lesions in harbour porpoises, as well as samples from unaffected skin. Work published recently, which included some of the authors on the current submission, documented attack by grey seals on harbour porpoises and the isolation of *Neisseria*

animalis from lesions following these attacks, which ultimately resulted in the death of the animals.

The current ms provides evidence that most samples from porpoise lesions contained microbiomes with organisms likely derived from seals during bite attacks, including four samples that were highly similar to grey seal oral flora samples. Many of these organisms are noted as potentially pathogenic and zoonotic. Five samples collected from the unaffected skin of porpoises, included as controls, were less similar to seal oral flora, thereby further supporting the transfer of organisms from the grey seals to porpoises during attacks. One caveat in this study, which the authors do point out, is that the oral flora were sampled from juvenile seals, whereas attacks on porpoises are associated with adult seals.

Interestingly, the authors did not identify *N. animalis* from any of the porpoise lesions sampled in this paper, however, it was only detected from the mouths of four grey seals and only in low or moderate numbers. The previous paper reporting its isolation from porpoise lesions did include Dutch animals, however, it is not stated whether these animals were included in this study. This needs clarified and warrants a little more discussion.

Bacterial transfer was apparent in 4/31 porpoise lesions but 3 of these appear to be from the same porpoise (UT1514). Please also present as number of porpoises. Based on other work, is it likely that a single seal would be involved in each attack – if so add this information in text?

I did not find any comment on whether there were differences based on the site of the individual lesions. Figure 2 lists lesion sites as numbers, could this be coded such that lesion site is clearer and a comment added in the text. It may also enable easier comparison between acute and chronic lesions.

Further comments, mostly typographical include:

The finding of large numbers of *Porphyromonas* as well as other anaerobes such as *Fusobacterium* emphasises the importance of anaerobe culture for diagnostic workers. Could a comment with this advice be added.

Other points mostly typographical:

General – when writing about skin samples from five controls, it would read better to specify these as unaffected skin samples. Likewise when other sample types is used when it appears that it is only unaffected skin samples that are being affected to.

90 – change ‘one juvenile was not having teeth yet’ to ‘Teeth had not erupted on one juvenile seal’

105 – typo – ‘basepare’ should be ‘basepair’

Lines 169–173 – I find the use of two ID numbers a little confusing for these key samples. This occurs elsewhere including the tables and figures, e.g. BV-16323-3 is included in Fig 2 but is absent from Table 1.

Line 191 – change ‘particular in grey seal’ to ‘in particular in grey seals’

204 – Typo - *Streptobacillus*

227-8 – provide number of lesions/ animals?

319 – this sentence on *Fusobacterium* does not follow well from the previous sentence which is on sea water and sediment.

326 – typo ‘grey greys’

325-32 – would it be appropriate to add a reference from other mammalian hosts which have compared oral microbiomes at different ages?

327 – typo – ‘aging’

334 – change – ‘not sure’ to ‘unclear’

339 – change ‘lesion’ to ‘lesions’

350 – delete first ‘be’ before ‘also’

354-355 – *Brucella* has been reported from porpoise skin lesions previously: *Veterinary Microbiology* 90:563-580

370-2 M phocicerebrale has been reported from porpoise lungs previously: Journal of Wildlife Diseases 47:206-211. Given the typical host-adapted nature of mycoplasmas and its introduction to humans in seal finger cases, this report provides a possible reason for its finding in another host.

388-92 – How feasible is it for a porpoise to lick different parts of its own skin? Is it possible that a fellow porpoise would lick the wound of an injured animal, thereby introducing further organisms to a fresh wound?

Lines 394-396 – The authors previous N. animaloris paper should be referenced here as additional support.

Reviewer: 2

Comments to the Author(s)

Please see attached file.

Author's Response to Decision Letter for (RSOS-192079.R0)

Appendix B.

RSOS-192079.R1 (Revision)

Review form: Reviewer 1 (Geoffrey Foster)

Is the manuscript scientifically sound in its present form?

Yes

Are the interpretations and conclusions justified by the results?

Yes

Is the language acceptable?

Yes

Do you have any ethical concerns with this paper?

No

Have you any concerns about statistical analyses in this paper?

No

Recommendation?

Accept as is

Comments to the Author(s)

I am satisfied that all comments have been dealt with in the amended ms.

Review form: Reviewer 2

Is the manuscript scientifically sound in its present form?

Yes

Are the interpretations and conclusions justified by the results?

Yes

Is the language acceptable?

Yes

Do you have any ethical concerns with this paper?

No

Have you any concerns about statistical analyses in this paper?

No

Recommendation?

Accept with minor revision (please list in comments)

Comments to the Author(s)

This interesting article is improved and clarified important areas. My comments are minor and mostly grammatical.

Line 134 and 135 - perhaps indicate that the skin is intact skin since you have done this elsewhere in the article

Line 263 - shouldn't the porpoise be porpoises in both places.

Line 289-293 - the revision to this sentence seems confusing and it is a very long and complex sentence

Line 313 - I believe it might read better to say "at an individual level"

Line 348-349 - it seems repetitive to say that there are no teeth yet and lack of teeth in the same sentence

Line 391 - I'm not sure you need the word "Particularly" in this sentence now.

Decision letter (RSOS-192079.R1)

10-Mar-2020

Dear Dr Gilbert:

On behalf of the Editors, I am pleased to inform you that your Manuscript RSOS-192079.R1 entitled "After the bite: bacterial transmission from grey seals (*Halichoerus grypus*) to harbour porpoises (*Phocoena phocoena*)" has been accepted for publication in Royal Society Open Science subject to minor revision in accordance with the referee suggestions. Please find the referees' comments at the end of this email.

The reviewers and Subject Editor have recommended publication, but also suggest some minor

revisions to your manuscript. Therefore, I invite you to respond to the comments and revise your manuscript.

- Ethics statement

- Data accessibility

<http://datadryad.org/submit?journalID=RSOS&manu=RSOS-192079.R1>

- Competing interests

- Authors' contributions

- Acknowledgements

- Funding statement

Because the schedule for publication is very tight, it is a condition of publication that you submit the revised version of your manuscript before 19-Mar-2020. Please note that the revision deadline will expire at 00.00am on this date. If you do not think you will be able to meet this date please let me know immediately.

on behalf of Dr Denise Greig (Associate Editor) and Kevin Padian (Subject Editor)
openscience@royalsociety.org

Associate Editor Comments to Author (Dr Denise Greig):

Thank you very much for your revision and addressing reviewer concerns. My remaining comments, like reviewer #2, are mostly grammatical:

Line 274-277. To address reviewer 2 concerns, this sentence could be changed to "Bacterial transfer from seals to harbour porpoises by biting is highly likely based on these results: in addition to the four harbour porpoise skin lesion microbiomes that were similar to seal oral microbiomes, many other skin lesion microbiomes contained OTUs typically associated with seals or the oral cavity in general."

Line 296, change "also" to "even"

Line 308, delete "being"

Line 320-321. Change "despite that the individual animals were all wild and from different locations..." to "despite collection from different locations".

Lines 325-329. Change "Nevertheless, based on the high similarity of the oral microbiomes, while being from two different species, the previous close contact between juvenile and maternal animals, facilitating bacterial transfer, and the presence of many bacterial species which are also associated with adult seals, many components of the adult oral microbiome are likely already present in the juvenile oral microbiome." to

"Nevertheless, the presence of many bacterial species which previously have been associated with adult seals suggests that many components of the adult oral microbiome were already present in the juvenile oral microbiome."

Line 333-337. I think you can delete this entire paragraph as I do not see that you made any general assumptions about different sites on the body.

Line 379. Change "was" to "were"

Line 381. Delete "as well"

Line 389. Add a "," after "included"

Line 390. Delete "mostly"

Reviewer comments to Author:

Reviewer: 1

Comments to the Author(s)

I am satisfied that all comments have been dealt with in the amended ms.

Reviewer: 2

Comments to the Author(s)

This interesting article is improved and clarified important areas. My comments are minor and mostly grammatical.

Line 134 and 135 - perhaps indicate that the skin is intact skin since you have done this elsewhere in the article

Line 263 - shouldn't the porpoise be porpoises in both places.

Line 289-293 - the revision to this sentence seems confusing and it is a very long and complex sentence

Line 313 - I believe it might read better to say "at an individual level"

Line 348-349 - it seems repetitive to say that there are no teeth yet and lack of teeth in the same sentence

Line 391 - I'm not sure you need the word "Particularly" in this sentence now.

Author's Response to Decision Letter for (RSOS-192079.R1)

See Appendix C.

Decision letter (RSOS-192079.R2)

24-Mar-2020

Dear Dr Gilbert,

It is a pleasure to accept your manuscript entitled "After the bite: bacterial transmission from grey seals (*Halichoerus grypus*) to harbour porpoises (*Phocoena phocoena*)" in its current form for publication in Royal Society Open Science.

on behalf of Dr Denise Greig (Associate Editor) and Kevin Padian (Subject Editor)
openscience@royalsociety.org

Appendix A

Royal Society of Open Science RSOS-192079

This is an interesting study examining the contribution of seal oral cavity bacteria to wounds found on porpoises using a 16S metagenomic approach. Overall it is a sound paper although a few conclusions are perhaps overstated. One of my major concerns is that some of the samples were taken from dead animals and may reflect some degree of post-mortem overgrowth. Although the authors identify this as a potential bias I don't think this possibility is emphasized enough. Also, there seems to be a fair amount of "discussion" within the results section. My specific comments follow.

Line 33 – the authors state that their study “provide insights in the zoonotic potential of bacteria present...” I think this is a misstatement. They show evidence of zoonotic bacterial presence but not the “zoonotic potential.”

Line 36 – potential pathogens for who? Dolphins, humans? Perhaps the earlier statement about potential zoonotic pathogens would better fit here or some modification of it?

Line 72-73 – The authors state they looked at acute and chronic wounds but don't provide much in the paper itself to distinguish. Perhaps somewhere here or in the methods the terms acute and chronic should be further defined. I believe that is done elsewhere but stating this earlier would be helpful for clarity

Line 74 – again, you did not look at the zoonotic potential, you identified bacteria genera, and some species, that might be zoonotic.

Line 80 – please define shortly. In my experience working with wildlife workers who sample stranded animals, recent or “fresh” might be several days post mortem.

Line 87 – please eliminate the “been” in “had been tested”

Line 89 – perhaps this should be tooth base? Also, where in the oral cavity were these samples taken, rostral, caudal, buccal/lingual location. This has been shown to matter in periodontal metagenomic studies

Line 90 – perhaps “did not have” would be better than “was not having”

Line 94 – should this be controls rather than control?

Line 95 – how long were samples stored before DNA extraction and were they stored in any solution?

Line 119 – There seems to be some discussion items in the results that I will try to point out some instances but in general I would prefer to see these in the discussion rather than the results

Line 134-136 – Isn't there a way to statistically analyze these differences in beta diversity? In figure 2 the authors show a 95% confidence region so it seems that this might be available

Line 145 – It seems that the word “samples” might be appropriate after “common seal”. Otherwise you might say porpoises and seals (plural)

Line 167 – Please specify that the samples that showed overlap were from the porpoises if this is the case. If it isn't then please clarify

Line 169-180 – it might be informative to indicate a few of these samples corresponding to those mentioned in this paragraph in figure 2 to clarify and improve understanding of the figure and this paragraph.

Line 179 – the authors state that HG16-014 was clearly distinct, but it isn't clear from figure 2 which sample this is.

Line 191 – perhaps rephrase to in grey seals or grey seal samples.

Line 194-195 – this seems like discussion. Also, the way this sentence is written it sounds like all seals that have PDV die of *Strep phocae*. Yes, it is a cause of death but not always.

Line 195 – This may be an editor comment but I don't think it is appropriate to start a sentence with an abbreviation e.g. S. Also consider rephrasing to *Streptococcus phocae* has been isolated from other....

Line 204 – it should be *Streptobacillus* not *Steptobacillus*

Line 206-209 – Not sure this is necessary and definitely belongs in the discussion

Line 228-230 – also belongs in the discussion

Line 232 – Potential pathogens, do you mean potential zoonotic pathogens?

Line 268 – same comment, zoonotic pathogens

Line 276 – please consider changing “indicating” to “consistent with”

Line 278 - *Streptobacillus*

Line 280 – this seems repetitive

Line 309-310 – the idea of post-mortem changes seems to be an afterthought. I think this potentially confounding concern should be brought forward more prominently particularly since you say on lines 334-336 that dead or alive might have a big influence.

Line 319 – you might bring out the idea of the requirement for many *Fusobacteria* to live in an anaerobic environment.

Line 327 – please provide a reference for changes with aging.

Line 330-332 – How do you know that the microbiomes won't change much? This seems contradictory to other statements earlier in the discussion and not supported by the literature.

Line 334 – dead rather than death.

Line 334-336 – I agree that the live/dead status may have greatly affected your results and may influence your conclusions. If this is the case, how did the authors come to such concrete conclusions?

Line 347-350 – some of the conclusions about zoonotic bacteria being present seem to be based on identification of genera not species. Is this appropriate?

Line 351 – What kind of biting incidents? This is phrased as if the humans are doing the biting but I think you mean seals and porpoises

Line 375-376 – again there is a conclusion that OTUs identified are zoonotic based only on genus

Line 338-392 – the authors don't address that much of the microbiome that isn't seal may have arisen from normal skin flora

Line 394 – potential zoonotic pathogens; consider changing “propensity to” to “possibility for”

Tables and Figures:

Table 1 appears 2 times in the pdf I downloaded. Is there a difference between them? If so, perhaps they could be combined?

Appendix B

We thank all reviewers for their critical review of the manuscript. Please find our response to the reviewers below.

Associate Editor Comments to Author (Dr Denise Greig):

This is an interesting manuscript providing additional evidence around newly described predatory behavior by grey seals on harbour porpoises. I agree with the reviewer comments, particularly around sample type, sample location, acute versus chronic lesions, and samples collected from live versus dead porpoises, as well as labelling the animals and their specimens clearly and consistently in the text and tables and figures.

I have a few additional questions detailed below:

Line 86 – when you say the seals are “juveniles”, how young do you mean? Are they pre-weaned? Recently weaned? Foraging on adult prey items? (clearly the one with unerupted teeth was pre-weaned).

All seals had an estimated age of 3 days up to 7 months. For clarity, this information has been added to the manuscript.

And a follow-up question is how does an oral microbiome develop: does it develop in utero, come from adult skin during suckling, prey items?

An oral microbiome is formed during and after birth, and is influenced by various factors including suckling, other physical interactions with mother and other conspecifics, and from the environment, including feed.

Line 111. Thank you for supplying your R code. It would be great if there was a little more annotation of the various commands.

More annotation has been added.

Line 182. Is the 12.9% (4/31)...4 out of 31 porpoises? Or porpoise lesions?

This would be the proportion of lesion samples. The text has been modified to clarify.

Line 201-202. Is this result visible in Figure 2? Is there a way to label it?

Figure 2 depicts the “overall similarity” of the complete microbiomes, which is composed of all (thousands of) individual OTUs. Showing individual OTUs in figure 2 will make the figure less clear. However, the occurrence of particular OTUs can easily be found in Table S1. In addition, we added figure S1, which includes sample names for easier cross-reference.

Line 274. Do you mean “sympatric” instead of “synoptic”?

Indeed sympatric may be more appropriate here. It has been modified in the manuscript.

Lines 358-361. I agree that *Brucella* is of zoonotic concern for people, but I think all the documented human infections have come from the dolphin specific species, right? It might be worth noting that since we don't know which species you detected in the seals or porpoises.

*The exact origin is not clear in all human cases. While in one case a cetacean origin was most likely, in other cases strains most closely related to *B. pinnipedialis* were involved, making a seal origin plausible.*

Reviewer comments to Author:

Reviewer: 1

Comments to the Author(s)

The authors present a microbiome study of the oral flora of seals and harbour porpoises and compare these with the flora of skin bite lesions in harbour porpoises, as well as samples from unaffected skin. Work published recently, which included some of the authors on the current submission, documented attack by grey seals on harbour porpoises and the isolation of *Neisseria animaloris* from lesions following these attacks, which ultimately resulted in the death of the animals.

The current ms provides evidence that most samples from porpoise lesions contained microbiomes with organisms likely derived from seals during bite attacks, including four samples that were highly similar to grey seal oral flora samples. Many of these organisms are noted as potentially pathogenic and zoonotic. Five samples collected from the unaffected skin of porpoises, included as controls, were less similar to seal oral flora, thereby further supporting the transfer of organisms from the grey seals to porpoises during attacks. One caveat in this study, which the authors do point out, is that the oral flora were sampled from juvenile seals, whereas attacks on porpoises are associated with adult seals.

Interestingly, the authors did not identify *N. animaloris* from any of the porpoise lesions sampled in this paper, however, it was only detected from the mouths of four grey seals and only in low or moderate numbers. The previous paper reporting its isolation from porpoise lesions did include Dutch animals, however, it is not stated whether these animals were included in this study. This needs clarified and warrants a little more discussion.

These animals have not been included. We have clarified this in the manuscript.

Bacterial transfer was apparent in 4/31 porpoise lesions but 3 of these appear to be from the same porpoise (UT1514). Please also present as number of porpoises.

Two lesions were from one animal (UT1514), the other two were from two different animals. It has also been presented as number of animals in the manuscript.

Based on other work, is it likely that a single seal would be involved in each attack – if so add this information in text?

The three porpoises with signs of bacterial transfer were from three different locations, making attacks by one seal not impossible, but less likely.

I did not find any comment on whether there were differences based on the site of the individual lesions. Figure 2 lists lesion sites as numbers, could this be coded such that lesion site is clearer and a comment added in the text. It may also enable easier comparison between acute and chronic lesions.

Differences between the sites of the individual lesions were not expected and have not been assessed in this study. All lesion sites were treated similarly. We added figure S1, which includes sample names for easier cross-reference. In this figure also a distinction is made between acute and chronic lesions.

Further comments, mostly typographical include:

The finding of large numbers of *Porphyromonas* as well as other anaerobes such as *Fusobacterium* emphasises the importance of anaerobe culture for diagnostic workers. Could a comment with this advice be added.

This has been added to the manuscript.

Other points mostly typographical:

General – when writing about skin samples from five controls, it would read better to specify these as unaffected skin samples. Likewise when other sample types is used when it appears that it is only unaffected skin samples that are being affected to.

This has been clarified throughout the manuscript.

90 – change ‘one juvenile was not having teeth yet’ to ‘Teeth had not erupted on one juvenile seal’

Changed.

105 – typo – ‘basepare’ should be ‘basepair’

Corrected.

Lines 169–173 – I find the use of two ID numbers a little confusing for these key samples. This occurs elsewhere including the tables and figures, e.g. BV-16323-3 is included in Fig 2 but is absent from Table 1.

For these samples the case/animal numbers (starting with UT) are included for clarity and cross-reference with Table 1. Sample numbers and associated case numbers are both shown in Table S1.

Line 191 – change ‘particular in grey seal’ to ‘in particular in grey seals’

Changed.

204 – Typo - Streptobacillus

Corrected.

227-8 – provide number of lesions/animals?

Added.

319 – this sentence on Fusobacterium does not follow well from the previous sentence which is on sea water and sediment.

This sentence has been moved.

326 – typo ‘grey greys’

Corrected.

325-32 – would it be appropriate to add a reference from other mammalian hosts which have compared oral microbiomes at different ages?

A reference has been added.

327 – typo – ‘aging’

Corrected.

334 – change – ‘not sure’ to ‘unclear’

Changed.

339 – change ‘lesion’ to ‘lesions’

Changed.

350 – delete first ‘be’ before ‘also’

Deleted.

354-355 – Brucella has been reported from porpoise skin lesions previously: Veterinary Microbiology 90:563-580

This reference has been added.

370-2 M phocicerebrale has been reported from porpoise lungs previously: Journal of Wildlife Diseases 47:206-211. Given the typical host-adapted nature of mycoplasmas and its introduction to humans in seal finger cases, this report provides a possible reason for its finding in another host.

This reference and its context have been added.

388-92 – How feasible is it for a porpoise to lick different parts of its own skin? Is it possible that a fellow porpoise would lick the wound of an injured animal, thereby introducing further organisms to a fresh wound?

Those options are also plausible. Given the normally gastric niche of Helicobacter cetorum, contamination may be most probable.

Lines 394-396 – The authors previous N. animaloris paper should be referenced here as additional support.

The reference has been added.

Reviewer: 2

Comments to the Author(s)

Royal Society of Open Science RSOS-192079

This is an interesting study examining the contribution of seal oral cavity bacteria to wounds found on porpoises using a 16S metagenomic approach. Overall it is a sound paper although a few conclusions are perhaps overstated. One of my major concerns is that some of the samples were taken from dead animals and may reflect some degree of post-mortem overgrowth. Although the authors identify this as a potential bias I don't think this possibility is emphasized enough. Also, there seems to be a fair amount of "discussion" within the results section. My specific comments follow.

Line 33 – the authors state that their study “provide insights in the zoonotic potential of bacteria present...” I think this is a misstatement. They show evidence of zoonotic bacterial presence but not the “zoonotic potential.”

The sentence has been modified accordingly.

Line 36 – potential pathogens for who? Dolphins, humans? Perhaps the earlier statement about potential zoonotic pathogens would better fit here or some modification of it?

Potential pathogens in general, but specifically for porpoises and humans.

Line 72-73 – The authors state they looked at acute and chronic wounds but don't provide much in the paper itself to distinguish. Perhaps somewhere here or in the methods the terms acute and chronic should be further defined. I believe that is done elsewhere but stating this earlier would be helpful for clarity

This has been further clarified in the methods.

Line 74 – again, you did not look at the zoonotic potential, you identified bacteria genera, and some species, that might be zoonotic.

Adapted.

Line 80 – please define shortly. In my experience working with wildlife workers who sample stranded animals, recent or “fresh” might be several days post mortem.

This has been defined in the manuscript.

Line 87 – please eliminate the “been” in “had been tested”

Done.

Line 89 – perhaps this should be tooth base? Also, where in the oral cavity were these samples taken, rostral, caudal, buccal/lingual location. This has been shown to matter in periodontal metagenomic studies

Adapted. The samples were taken at all the mentioned locations to get a complete overview of the bacteria present.

Line 90 – perhaps “did not have” would be better than “was not having”

Adapted.

Line 94 – should this be controls rather than control?

Adapted.

Line 95 – how long were samples stored before DNA extraction and were they stored in any solution?

Samples were stored without any solution for a couple of days up to a couple of weeks.

Line 119 – There seems to be some discussion items in the results that I will try to point out some instances but in general I would prefer to see these in the discussion rather than the results

This has been adapted throughout the manuscript where possible.

Line 134-136 – Isn't there a way to statistically analyze these differences in beta diversity? In figure 2 the authors show a 95% confidence region so it seems that this might be available

It was not feasible to analyse the differences in beta diversity in a reliable way, therefore we refrained from including this in the manuscript.

Line 145 – It seems that the word “samples” might be appropriate after “common seal”. Otherwise you might say porpoises and seals (plural)

Changed to plural.

Line 167 – Please specify that the samples that showed overlap were from the porpoises if this is the case. If it isn't then please clarify

This has been specified in the manuscript.

Line 169-180 – it might be informative to indicate a few of these samples corresponding to those mentioned in this paragraph in figure 2 to clarify and improve understanding of the figure and this paragraph.

We added figure S1, which includes sample names for easier cross-reference.

Line 179 – the authors state that HG16-014 was clearly distinct, but it isn't clear from figure 2 which sample this is.

The added figure S1 should clarify this.

Line 191 – perhaps rephrase to in grey seals or grey seal samples.

This has been rephrased accordingly.

Line 194-195 – this seems like discussion. Also, the way this sentence is written it sounds like all seals that have PDV die of *Strep phocae*. Yes, it is a cause of death but not always.

This has been moved and adapted.

Line 195 – This may be an editor comment but I don't think it is appropriate to start a sentence with an abbreviation e.g. S. Also consider rephrasing to *Streptococcus phocae* has been isolated from other....

Adapted accordingly.

Line 204 – it should be *Streptobacillus* not *Steptobacillus*

Corrected.

Line 206-209 – Not sure this is necessary and definitely belongs in the discussion

Moved to discussion.

Line 228-230 – also belongs in the discussion

Moved to discussion.

Line 232 – Potential pathogens, do you mean potential zoonotic pathogens?

Here we mean pathogens in a broader sense.

Line 268 – same comment, zoonotic pathogens

Also here we mean pathogens in a broader sense

Line 276 – please consider changing “indicating” to “consistent with”

Changed.

Line 278 – Streptobacillus

Corrected.

Line 280 – this seems repetitive

This has been rephrased.

Line 309-310 – the idea of post-mortem changes seems to be an afterthought. I think this potentially confounding concern should be brought forward more prominently particularly since you say on lines 334-336 that dead or alive might have a big influence.

This has been changed and should be more prominent now.

Line 319 – you might bring out the idea of the requirement for many Fusobacteria to live in an anaerobic environment.

This has been moved.

Line 327 – please provide a reference for changes with aging.

A reference has been added.

Line 330-332 – How do you know that the microbiomes won't change much? This seems contradictory to other statements earlier in the discussion and not supported by the literature.

We can't know exactly based on this data, but due to the reasons mentioned in the manuscript we expect that many components of the adult microbiome are already present. Although relative abundance may change over time, many bacterial species previously identified in adult animals have been identified in these juvenile animals as well. This sentence has been rephrased for clarification.

Line 334 – dead rather than death.

Corrected.

Line 334-336 – I agree that the live/dead status may have greatly affected your results and may influence your conclusions. If this is the case, how did the authors come to such concrete conclusions?

Many of the bacteria identified in the lesions are associated with seals and/or oral cavities. These bacteria are also commonly present in live seals.

Line 347-350 – some of the conclusions about zoonotic bacteria being present seem to be based on identification of genera not species. Is this appropriate?

We believe this is appropriate. Many genera contain multiple species which may be pathogenic and/or zoonotic. When considered relevant (e.g. high presence), these have been highlighted.

Line 351 – What kind of biting incidents? This is phrased as if the humans are doing the biting but I think you mean seals and porpoises

The sentence has been rephrased.

Line 375-376 – again there is a conclusion that OTUs identified are zoonotic based only on genus

This is not meant to be read as a conclusion, merely as a precaution. As many members of the Mycoplasma genus are considered pathogenic, and due to the prominent role of this genus in seal-human zoonotic events, we believe this precaution is appropriate.

Line 338-392 – the authors don't address that much of the microbiome that isn't seal may have arisen from normal skin flora

This has been addressed in both results and discussion (e.g. lines 304-305). Indeed focus of the manuscript is on the bacterial transfer from seal to harbour porpoise lesion.

Line 394 – potential zoonotic pathogens; consider changing “propensity to” to “possibility for”

Adapted.

Tables and Figures:

Table 1 appears 2 times in the pdf I downloaded. Is there a difference between them? If so, perhaps they could be combined?

These tables are duplicates.

Appendix C

We thank all reviewers for their critical review of the manuscript. Please find our response to the reviewers below.

Associate Editor Comments to Author (Dr Denise Greig):

Thank you very much for your revision and addressing reviewer concerns. My remaining comments, like reviewer #2, are mostly grammatical:

Line 274-277. To address reviewer 2 concerns, this sentence could be changed to “Bacterial transfer from seals to harbour porpoises by biting is highly likely based on these results: in addition to the four harbour porpoise skin lesion microbiomes that were similar to seal oral microbiomes, many other skin lesion microbiomes contained OTUs typically associated with seals or the oral cavity in general.

The sentence has been changed accordingly.

Line 296, change “also” to “even”

Changed.

Line 308, delete “being”

Deleted.

Line 320-321. Change “despite that the individual animals were all wild and from different locations...” to “despite collection from different locations”.

Changed.

Lines 325-329. Change “Nevertheless, based on the high similarity of the oral microbiomes, while being from two different species, the previous close contact between juvenile and maternal animals, facilitating bacterial transfer, and the presence of many bacterial species which are also associated with adult seals, many components of the adult oral microbiome are likely already present in the juvenile oral microbiome.” to

“Nevertheless, the presence of many bacterial species which previously have been associated with adult seals suggests that many components of the adult oral microbiome were already present in the juvenile oral microbiome.”

Changed.

Line 333-337. I think you can delete this entire paragraph as I do not see that you made any general assumptions about different sites on the body.

This paragraph has been deleted.

Line 379. Change “was” to “were”

Changed.

Line 381. Delete “as well”

Deleted.

Line 389. Add a “,” after “included”

Added.

Line 390. Delete "mostly"

Deleted.

Reviewer comments to Author:

Reviewer: 1

Comments to the Author(s)

I am satisfied that all comments have been dealt with in the amended ms.

Reviewer: 2

Comments to the Author(s)

This interesting article is improved and clarified important areas. My comments are minor and mostly grammatical.

Line 134 and 135 - perhaps indicate that the skin is intact skin since you have done this elsewhere in the article

This has been indicated.

Line 263 - shouldn't the porpoise be porpoises in both places.

Adapted.

Line 289-293 - the revision to this sentence seems confusing and it is a very long and complex sentence

This sentence has been shortened and simplified.

Line 313 - I believe it might read better to say "at an individual level"

Adapted.

Line 348-349 - it seems repetitive to say that there are no teeth yet and lack of teeth in the same sentence

This sentence has been modified.

Line 391 - I'm not sure you need the word "Particularly" in this sentence now.

This word has been deleted.